# Study on Deposition Characteristics of Microparticles in Terminal Pulmonary Acini by IB–LBM

**DOI:** 10.3390/mi12080957

**Published:** 2021-08-13

**Authors:** Du-Chang Xu, Yu-Xiao Luo, Yuan-Qing Xu

**Affiliations:** 1School of Life Science, Beijing Institute of Technology, Beijing 100081, China; bitxdc@163.com; 2University Medical Center Göttingen, University of Göttingen, 37075 Göttingen, Germany; yuxiao.luo@stud.uni-goettingen.de

**Keywords:** microparticles deposition, terminal pulmonary acini, IB–LBM

## Abstract

As an indicator of health risk, the deposition of microparticles in terminal pulmonary acini is of great significance in the medical field. To control particulate pollution and optimize aerosol delivery, it is necessary to perform an in-depth study of the microparticle deposition in terminal pulmonary acini; however, little research has been done on this topic. This paper proposes a respiratory movement model of terminal pulmonary acini using an immersed boundary–lattice Boltzmann method. In addition, we explored the effect of gravity direction, respiratory rate, microparticle diameter, and other parameters on the microparticles deposition process and distribution, under the airflow in the acinar wall. It was found that the deposition of microparticles is sensitive to gravity direction, and the growth of the respiratory rate increases the rate of microparticle migration and deposition. It was observed that the gravity effect is enhanced by increasing the diameter of microparticles, causing a high deposition and dispersion rate. The study reveals the dynamic correlation between the respiration process and the movement of microparticles, which is of reference value to figure out the pathogenicity mechanism of inhalable particles and to optimize the aerosol delivery.

## 1. Introduction

There are 274–790 million (480 million on average) alveoli in human lungs that constitute a large surface area of 70–75 m^2^ for gas exchange between the inhaled air and the blood [1]. An acinus is defined as the blind-ending parenchymal unit beginning with a transitional bronchiole. Within an acinus, all airways (alveolar ducts and respiratory bronchioles) have alveoli attached to their walls, which are the primary sites of oxygen and carbon dioxide exchange [2]. The acini are composed of three distinct structures: respiratory bronchioles, alveolar ducts, and alveolar sacs [3]. Respiratory bronchioles split into alveolar ducts, which, further, branch into alveolar sacs, terminally [4]. Alveolar sacs are of great importance to pulmonary function [5].

Some respiratory diseases affect pulmonary acini, where microparticles are inhaled and deposited, causing infection, inflammation, and even tumor formation [6]. Moreover, certain viruses and bacteria are responsible for lung diseases by invading pulmonary acini directly, such as inhalational anthrax, which results from the deposition of spore-bearing microparticles of 1 to 5 μm into the pulmonary acini [7]. Knowing the deposition of microparticles in terminal pulmonary acini helps predict and prevent respiratory diseases.

Injection is the most common method of drug delivery. Substantial progress has been made in recent years regarding nanoparticle and microparticle delivery in blood circulation. It is found that ellipsoidal particles with submicron shapes are more liable to adhere to the wall surface [8,9]. Drug particles need to overcome a series of complex biological obstacles to reach biological targets in blood circulation. By comparison, pulmonary aerosol delivery is more efficient. As an emerging method, pulmonary aerosol delivery is significant to treat respiratory diseases. It is an effective and non-invasive pathway for therapeutic delivery, especially for drugs that cannot be delivered orally due to degradation in the digestive system [10]. Alveolar sacs are the functional targets for inhalation drug deposition and absorption, where, for example, inhaled insulin and some other drugs are absorbed. The medicines used for distal lungs, such as pentamidine depositing in upper airways, may cause significant side effects [11]. Thus, the size and the way of inhalation of atomized droplets should be well designed.

Based on the reasons above, it is of great scientific and clinical significance to study the deposition of microparticles in pulmonary acini. In some studies, a hemispherical shell structure and a cylindrical pipe were used to simulate the alveolus and bronchiole, respectively, and a preset airflow was designed to drive microparticles. The application of this rigid wall structure helps to study the movement and deposition of microparticles in pulmonary acini [12,13,14]. Compared to the rigid wall, the simulation results of using alveolus with rhythmic wall motion model is more similar to the physiological situation, and the deposition characteristics of microparticles are more accurate, accordingly. Tsuda et al. [15] simulated a two-dimensional axisymmetric alveolar structure, expanding and contracting sinusoidally, and the ratio of alveolar flow to ductal flow was found to significantly influence the flow patterns in the alveolus, resulting in changes to the characteristics of alveolus movement, such as alteration in the area of circulatory flow within the alveolus during respiration and the dispersion of microparticles [16]. In general, the movement of the alveolar wall significantly affects the airflow inside the alveolus [17,18,19]. Meanwhile, the change in flow patterns will alter the movement and distribution of microparticles [20,21]. In addition, much attention has been paid to flow patterns and the movement of microparticles in the bronchial-alveolar system. Tsuda et al. [22] studied the multi-alveolar structure in pulmonary acini, and they found that the deposition of microparticles is strongly affected by the alveolar structural characteristics. Additionally, the complexity of airway structure in pulmonary acini impacts the deposition of microparticles [23,24].

The deposition of microparticles smaller than 20 μm has been generally studied [25], and the effect of Brownian movement to the microparticles, whose diameters are larger than 1 μm, are negligible. The movement and migration of microparticles is mainly controlled by aerodynamics and gravity [26]. Acinar flow shows the characteristics of chaotic flow [27]. Considering the inertial effect of microparticles, it is necessary to propose a model that can fully predict the movement.

So far, a number of studies have been conducted on the deposition of microparticles in the respiratory bronchioles and alveolar ducts. However, the studies of deposition in terminal pulmonary acini are still scarce, and the corresponding rules of deposition need to be further explored. In addition, the airflow in terminal acini is driven entirely by the expansion and contraction of the pulmonary acinar wall, in which the motion of microparticles is different from that in alveolar ducts. The dynamic transport process of microparticles in a closed sac is worth studying.

In order to study the movement and deposition of microparticles in terminal pulmonary acini, we built a respiratory movement model of terminal pulmonary acini using immersed boundary–lattice Boltzmann method (IB–LBM) [28,29,30,31,32,33,34], with an immersed boundary (IB) method describing the pulmonary acinar wall and the morphology of the microparticles. The lattice Boltzmann (LB) method has an advantage in dealing with flows in complex geometries [35,36]. We used an elastic mesh structure that is independent of the flow field to simulate the solid lung tissue, so that the elastic support of the terminal pulmonary acinus wall boundary was set. By changing the internal and external pressure of terminal pulmonary acini, we controlled the movement of the terminal wall to simulate the expansion and contraction process of terminal pulmonary acini. In addition, periodic expansion and contraction of the terminal wall was repeated to simulate multiple respiratory cycles. By building the model, we studied the movement and deposition rules of the microparticles in air flow under different respiratory rates, the diameter of microparticles, and other factors. This provides a deductive research method for evaluating the effects of haze and flying dust on the function of terminal lung acini. From the perspective of medical treatment, it provides a theoretical reference for improving the design of atomized drug administration.

The rest of the paper is organized as follows: Section 2 set up the model. Section 3 verifies the correctness of the model. In Section 4, the deposition characteristics of microparticles are studied in detail. The final summary and conclusion are given in Section 5.

## 2. Numerical Model

### 2.1. Alveolar Model

In this work, first, we built a model of terminal pulmonary acini that responded to the actual respiratory process (Figure 1). When pulmonary acini are under low pressure, the alveolar ducts are narrowed, and folds characterize the alveolar septums. When pulmonary acini are under high pressure, the alveolar ducts are wide, and the alveolar septums are stretched [37]. Our model can simulate this physiological process.

Each healthy human alveolus can be considered a small sphere, approximately 200 μm to 450 μm in diameter. Pulmonary acini are composed of many complex connections of alveoli and pulmonary ducts. In this study, part of the terminal pulmonary acini is selected for numerical modeling, which is the representative area of pulmonary acini. The diameter of the alveolar model is set as 400 μm, and the terminal pulmonary acini is composed of an alveolar duct of 200 μm diameter and surrounding alveoli. The alveolar membrane in this model is a continuous movable wall, whose position is from 0 to 1, as indicated. The respiratory flow curve of a healthy person approximates a sinusoidal curve [38]. The left side shows inlet and outlet of airflow, and the boundary condition is set as
(1)$$\rho_{1}=0.0015\sin\frac{t}{200000}+1$$
where $\rho_{1}$ is the inlet density and t is the time step. Air fluid density $\rho_{0}\;$in terminal pulmonary acini is a Newtonian fluid with density $\rho_{0}=1$ and kinematic viscosity nu=0.4. Fluid dynamics are governed by Equations (2) and (3).

Continuity equation:(2)∇⋅u=0

Navier–Stokes equation:(3)$$\rho \left(\frac{\partial u}{\partial t}+u\cdot \nabla u\right)+\nabla p=\mu \nabla^{2}u+f$$

The elastic structure of the alveolus is simulated by a series of unstructured grids [39,40].

These grids are directly connected to the alveolar boundary. To simulate the elastic support of the pulmonary tissue, the outer edges of these grids are set to be fixed, which move as the alveolar boundary deforms and generates elastic force at the alveolar boundary. These grids, which move as elastic force and bending force change, are set independently of the flow field. Elastic force is calculated by the equation [41]:(4)$$F_{S1}=K_{S1}\left(e^{2\Delta x}-e^{-\Delta x}\right)\vec{n}$$
where $K_{S1}$ is the elasticity coefficient of pulmonary tissue, Δx is the elastic elongation between adjacent grid points, and $\vec{n}$ is the unit direction vector. Bending force is computed by the equation:(5)$$F_{b}=K_{b}\frac{\partial^{4}X_{\left(s,t\right)}}{\partial s^{4}}$$
where $K_{b}$ is the bending stiffness and $\partial X_{\left(s,t\right)}/\partial s$ is the unit tangent vector. The elastic force of the alveolar membrane point is calculated by a neo-Hookean elastic membrane force equation [42]:(6)$$F_{S2}=K_{S2}\left(\lambda^{3/2}-\lambda^{-3/2}\right)\vec{n}$$
where $K_{S2}$ is the elasticity coefficient of the elastic membrane and λ is the stretch ratio of the distance between membrane points.

The alveolar movement is generated by changing the inlet density boundary condition, simulating the pressure change in pulmonary acini. The position of the alveolar membrane point changes by the flow field pressure, and the elastic force and bending force change accordingly. Eventually, the alveolus stops moving when reaching a new equilibrium.

### 2.2. Microparticles Model

We studied the movement of micron-sized particles so that gravity is taken into account and the effect of Brown movement is negligible.

Solid microparticles are simulated by the node-spring model, which consists of a central supporting point and some boundary points (Figure 2).

The elastic force between the adjacent grid points, bending force, and elastic force generated by the central support point are applied to each boundary point. Elastic force obeys Hooke’s Law, calculated by the equation:(7)$$F_{s3}=\frac{\partial }{\partial s}\left[K_{s3}\left(\left|\frac{\partial X_{\left(s,t\right)}}{\partial s}\right|-1\right)\frac{\partial X_{\left(s,t\right)}}{\partial s}\right]\;$$
where $K_{S3}$ is tensile rigidity and $\partial X_{\left(s,t\right)}/\partial s$ is the unit tangent vector. Bending force is calculated by Equation (5).

In addition to elastic force calculated by Equation (7), a central support point is acted on by gravity and inertia force. Gravity G is calculated by the equation:(8)G=mg,
where *m* is microparticle quality and *g* is gravity coefficient. Inertia force $F_{k}$ is calculated by the equation:(9)$$F_{k}=m_{s}\frac{\partial^{2}X_{\left(s,t\right)}}{\partial t^{2}}$$
where $m_{s}$ is inertia coefficient, which controls particle inertia. $X_{\left(s,t\right)}$ is the center point coordinate vector.

When a microparticle approaches the alveolar boundary, if the shortest distance between the center and boundary points is less than three times the microparticle radius, the adhesion mechanism of the alveolar boundary will be activated. Adhesion is simulated by a virtual spring, whose original length is twice the microparticle radius, which connects the central point to the nearest boundary point. Adhesion force $F_{c}$ is calculated by the equation:(10)$$F_{c}=\{\begin{array}{c} k_{c}\frac{l-l_{0}}{l_{0}}\vec{n}\;\;\;\;\;\;l\leq 3R\\ 0\;\;\;\;\;\;\;\;\;\;\;\;\;\;\;\;\;\;l>3R \end{array}$$
where $k_{c}$ is the elastic coefficient, $l_{0}$ is the expected riveting point distance, l is the actual distance, and R is microparticle radius. When microparticles are close to each other and collide, the repulsive force $F_{r}$ is set according to the momentum conservation rule. The activation distance is set to be twice the microparticle radius, and the limit proximity distance is not less than the microparticle radius. The repulsive force $F_{r}$ is calculated by the equation:(11)$$F_{r}=\{\begin{array}{c} k_{r}\frac{1}{{\left(l-R\right)}^{2}}\vec{n}\;\;\;\;\;R<l\leq 2R\\ 0\;\;\;\;\;\;\;\;\;\;\;\;\;\;\;l>2R \end{array}$$
where $k_{r}$ is the coefficient of repulsive force.

### 2.3. IB–LBM

When simulating particles in a flow, the lattice Boltzmann and Langevin dynamics (LB–LD) method can implement multi-scale particles movement simulation to study complex flows [43,44]. The IB–LBM can implement deformable boundaries and simulate the interaction between particles and the interaction between particles and the flow field. Additionally, at the micron scale, Brownian motion can be ignored, so, in this study, the IB–LBM is more suitable. The multi-relaxation time (MRT) model has a wider selection of parameters than the lattice Bhatnagar–Gross–Krook (LBGK) model, which is beneficial to propose a whole lung model in the future.

In this work, the discrete lattice Boltzmann method of the multi-relaxation time model is utilized to solve the Navier–Stokes equations. We can find the lattice Boltzmann equation:(12)$$g_{i}\left(x+e_{i}\Delta t,t+\Delta t\right)-g_{i}\left(x,t\right)=-S_{\alpha i}\left[g_{i}\left(x,t\right)-g_{i}^{\mathrm{eq}}\left(x,t\right)\right]$$
where $g_{i}\left(x,t\right)$ is the distribution function of velocity $e_{i}$ at position x and moment t,  Δt is the time step, S is collision matrix, and $g_{i}^{\mathrm{eq}}$ is the equilibrium distribution function.

Here, the 9 discrete velocities in 2 dimensions (D2Q9) model is used, and the nine velocities $e_{i}$ are given by the equation:(13)$$e_{i}=\{\begin{array}{c} (0,0)\;i=0\\ \left(\cos\frac{\pi \left(i-1\right)}{2},\;\sin\left(\frac{\pi \left(i-1\right)}{2}\right)\right)\frac{\Delta x}{\Delta t}\;i=1,\;2,\;3,\;4\\ \left(\cos\left(\frac{i-4.5}{2}\pi \right),\;\sin\left(\frac{i-4.5}{2}\pi \right)\right)\frac{\sqrt{2}\Delta x}{\Delta t}\;\;i=5,\;6,\;7,\;8 \end{array},$$
where Δx is the lattice spacing. The settings of $e_{i}$ enable a strategy to control the migratory directions within a time step. In Equation (12), $g_{i}^{\mathrm{eq}}$ is calculated by the equation:(14)$$g_{i}^{\mathrm{eq}}=\omega_{i}\rho \left[1+\frac{e_{i}\cdot u}{c_{s}^{2}}+\frac{uu:\left(e_{i}e_{i}-c_{s}^{2}I\right)}{2c_{s}^{4}}\right]$$
where u is fluid velocity and $\omega_{i}$ is the weight defined by $\omega_{0}=4/9$, $\omega_{i}=1/9$ for i=1−4, and $\omega_{i}=1/36$ for i=5−8. $c_{s}=\Delta x/\sqrt{3}\Delta t$ is the sound speed. According to Lallermand et al. [45], Equation (12) is transformed into the equation:(15)$$g_{i}\left(x+e_{i}\Delta t,t+\Delta t\right)-g_{i}\left(x,t\right)=-M^{-1}\hat{S}\left[{\hat{g}}_{i}\left(x,t\right)-{\hat{g}}_{i}^{\mathrm{eq}}\left(x,t\right)\right]$$
where M is the transformation matrix. $\hat{S}$ is the collision matrix of the moment space. In this model, the value of $\hat{S}$ is given by the equation [46]:(16)$$\hat{S}=diag\left[1,\;1.64,\;1.54,\;1,\;1.9,\;1,\;1.9,\;\frac{1}{\tau },\;\frac{1}{\tau }\right]$$
where τ is the relaxation time, calculated by the equation [47]:(17)$$\tau =\frac{\upsilon }{c_{s}^{2}\Delta t}+0.5$$
where υ is the kinematic viscosity in Navier–Stokes equations. Once the density distribution functions are known, the macroscopic fluid density, velocity, and pressure can be calculated by equations:(18)$$\;\rho =\underset{i}{\sum }g_{i}$$
(19)$$u=\frac{\sum_{i}e_{i}g_{i}+0.5f\Delta t}{\rho }$$
(20)$$P=\underset{i}{\sum }\rho c_{s}^{2}\;$$
where f is the vector of the body force density. The non-equilibrium extrapolation method is used to obtain the particle distribution function on the boundaries, which is marked with $g_{i,b}$, and calculated by the equation:(21)$$g_{i,b}\left(x,t\right)=g_{i,b}^{\mathrm{eq}}\left(x,t\right)+\left(g_{i,n}\left(x,t\right)-g_{i,n}^{\mathrm{eq}}\left(x,t\right)\right)\;$$
where $g_{i,n}$ represents the particle distribution function of the neighboring grid and $g_{i,b}^{\mathrm{eq}}$ and $g_{i,n}^{\mathrm{eq}}$ are, respectively, the equilibrium forms of $g_{i,b}$ and $g_{i,n}$.

The interaction between the fluid and particles is achieved by the immersed boundary method. In this method, Lagrangian force ***F*** is spread onto the collocated grid points near the boundary, which is calculated by the equation:(22)$$f\left(x,t\right)=\underset{\Gamma }{\int }F\left(s,t\right)D\left(x-X\right)ds$$
where D(x−X) is the Dirac delta function, calculated by the equation [48]:(23)$$D\left(x-X\right)=\overset{n}{\underset{i=1}{\prod }}\delta \left(x_{i}-X_{i}\right)\;$$
where n is the dimension and $\delta \left(x_{i}-X_{i}\right)$ is calculated by the Equation:(24)$$\delta \left(x_{i}-X_{i}\right)=\{\begin{array}{l} \frac{1}{8\Delta x}\left(3-\frac{2\left|r\right|}{\Delta x}+\sqrt{1+\frac{4\left|r\right|}{\Delta x}-4{\left(\left|\frac{r}{\Delta x}\right|\right)}^{2}}\;\right)\;\left|r\right|=\left|x-X\right|\leq \Delta x\\ \frac{1}{8\Delta x}\left(5-\frac{2\left|r\right|}{\Delta x}+\sqrt{-7+\frac{12\left|r\right|}{\Delta x}-4{\left(\left|\frac{r}{\Delta x}\right|\right)}^{2}}\right)\;\;\Delta x\leq \left|r\right|=\left|x-X\right|\leq 2\Delta x\\ 0\;\left|r\right|=\left|x-X\right|\geq 2\Delta x \end{array}$$

Then, the velocity U of the moving boundary X can be calculated by the equation:(25)$$U\left(s,t\right)=\int_{\Omega }u\left(x,t\right)D\left(x-X\right)dx$$

The position is updated by the equation:(26)$$\frac{\partial X}{\partial t}=U\left(s,t\right)$$

## 3. Validation

### 3.1. Grid Independence Validation

First, different support grids are set in the same alveolar model to compare the changes in alveolar expansion. By comparing the variation of alveolar expansion, the influence of the supporting grid on alveolar deformation can be identified, which provides a basis for the selection of the scale of the supporting grid. The outer boundary that supports the grid is a free flow boundary but a fixed border for the supporting grid with no displacement of nodes. The verification steps are as follows:In the initial state, the pressure difference between the alveolar inlet and the free flow boundary is 0. Calculate static pressure $P_{0}$ using the equation:(27)$$\;P_{0}=c_{s}^{2}\rho_{0}$$
where $\rho_{0}=1.00$ is initial density. The alveoli and supporting mesh are free of stress and deformation;Change the alveolar inlet pressure by the equation:(28)$$\;P_{1}=c_{s}^{2}\rho_{\mathrm{in}}$$
where $\rho_{\mathrm{in}}=1.02$ is inlet density. We can find the pressure difference ΔP by the equation:(29)$$\Delta P=c_{s}^{2}\left(\rho_{\mathrm{in}}-\rho_{0}\right)$$Record the volume of the alveolar expansion when the alveolar expansion reaches a steady state. Expansion ratio $D_{r}$ is calculated by the equation:(30)$$D_{r}=\frac{V-V_{0}}{V_{0}}\times 100\%$$
where V is the volume of the alveoli after expansion. $V_{0}$ is the volume of the alveoli without expansion. We use grid proportion $M_{r}$ to represent the variation of support grid, which is calculated by the equation:(31)$$M_{r}=\frac{n_{w}}{n_{m}},$$
where $n_{w}$ is the number of grid points and $n_{m}$ is the number of alveolar membrane points.

In this validation, we set $M_{r}$ as 64, 81, 100, and 121, respectively, with the spatial distribution of the corresponding supporting grid shown in Figure 3a–d. The blue areas surrounded by black lines represent this type of grid.

Changes in $M_{r}$ are shown in Figure 4. The expansion ratio $D_{r}$ of $M_{r}=100$ increases by 0.0065 over the expansion ratio $D_{r}$ of $M_{r}=81$. The expansion ratio $D_{r}$ of $M_{r}=121$ increases by 0.0035 over the expansion ratio $D_{r}$ of $M_{r}=100$. The results indicate that the influence of the supporting grid size on the alveolar deformation tends to converge with the increase in the grid proportion. The alveolar deformation, therefore, is considered not related to the size of the external supporting grid when the grid reaches a certain number.

### 3.2. Validation of Microparticle Inertia Expression Model

The present numerical method is validated by simulating the inertial migration of microparticles in a microchannel (Figure 5). The dimensions of the microchannel used in the simulations are [0, 2*l*] × [0, 3*l*], where *l* = 32 μm.

The central supporting point is initially located at 0.5*l* from the upper boundary and moves due to gravity G, which is proportional to the volume of the microparticle (the volume of the microparticle is represented by microparticle area in the two-dimensional case). When the microparticle moves to *l*, it continues to travel for some distance without gravity G. Microparticles with greater inertia move farther.

The test results are shown in Table 1 and Table 2. Table 1 indicates the sliding distances of microparticles with different diameters of *D* under the same inertia conditions. The larger the microparticle size, the greater the inertia force, so that the microparticle travels farther. Table 2 shows the sliding distances of the same microparticle under different inertial conditions. The sliding distance of the 5 μm particle, which is 11.37 μm, is the longest of the three sizes. Therefore, the 5 μm particles have the maximum inertia force. Moreover, as $m_{s}$ is 100, the sliding distance is the longest, at 16.08 μm. So, the larger the $m_{s}$ is, the greater the inertia force is, which makes the microparticle move farther.

The above results reveal that the microparticle model can effectively simulate the inertia of the microparticle in the movement process, which meets the requirements for the simulation of the inertia of solid microparticles in the lung.

## 4. Results and Discussion

### 4.1. Respiratory Deformation and Microparticles Movement in Pulmonary Acini

In the continuous respiratory cycle, pulmonary acini show the periodic movement of expansion and contraction, and microparticles entering pulmonary acini move and migrate by a variety of influencing factors. These factors include deformation amplitude, deformation velocity, microparticle mass, gravity direction, and mucus on the pulmonary acini surface. The Reynolds number (*Re*) of gas in pulmonary acini is very small (*Re* << 1) so that the airflow can be considered an isothermal incompressible laminar flow. We find the flow in pulmonary acini is relatively smooth with no obvious backflow area by simulation.

The deformation of pulmonary acini during one respiratory cycle is shown in Figure 6. In one respiratory cycle T, the times of inhalation and exhalation are each T/2. t∈(nT,nT+ T/2) is the inspiratory cycle. When t=nT+ T/2, pulmonary acini expand to their maximum volume at a constant positive inlet pressure. $t\in \left[\left(n+\frac{1}{2}\right)T,\left(n+1\right)T\right]$ is the expiration cycle; when t=(n+1)T, pulmonary acini shrink to their minimum at the constant negative inlet pressure.

To observe the movement and migration of particles in the process of pulmonary acini expansion and contraction, 160 microparticles of diameter 4 μm are set at the entrance (Figure 7, *t* = 0), which are subjected to airflow and gravity.

Figure 7 shows the movement and distribution of microparticles in ten respiratory cycles. The groups of microparticles are observed to spread gradually within acini over time, and some microparticles deposit on the acini membrane, which occurs with a certain regularity. This indicates that the model we constructed can effectively simulate the movement and distribution of multiple microparticles in pulmonary acini in a proper setting.

### 4.2. Movement and Distribution of Microparticles in Pulmonary Acini

When a microparticle approaches the inner surface of pulmonary acini, it will be captured by mucus, so that its position will not be affected by airflow or gravity. In order to include the trapping effect of mucilage on microparticles, a fixing mechanism is introduced to rivet the microparticles close to the surface of the mucus. A specific method is as follows:

When the shortest distance between the point of microparticle boundary and the point of pulmonary acinar boundary is smaller than the diameter of the microparticle, the center of the microparticle is set to connect with the corresponding border point of pulmonary acini by a spring with the original length of the microparticle diameter and spring with the original length of 0.5 μm longer than the length between the center of the microparticle and the corresponding border point of pulmonary acini.

This method better simulates the phenomenon of microparticle trapping on the inner surface. The elastic force is calculated by the equations:(32)$$F_{m1}=K_{s,m}\frac{s-D}{D}$$
(33)$$F_{m2}=K_{s,m}\frac{d-s}{d}$$
where $K_{s,m}$ is the corresponding elastic control coefficient, which keeps the change in the distance between the center of the microparticle and the related point of pulmonary acini boundary within 0.1D.

In addition, in order to indicate the number of microparticles captured by mucus, and describe the distribution of microparticles deposition, we counted the number of microparticles near the surface of pulmonary acini according to the following method.

We set the static distance between points on the surface of pulmonary acini as Δs (the distance between adjacent points when no expansion or contraction occurs), and counted the number of microparticles in the distance band with a width of 30 Δs inside the boundary of the pulmonary acini (the position of microparticles is represented by the center point). According to the point order of pulmonary acinar boundary, by starting from the left end, we normalized and located the boundary with a set of relative positions from 0 to 1, where 0 refers to the left end and 1 is the right end. The distribution of microparticle deposition on the inner surface of pulmonary acini can be obtained by sampling the number of microparticles with a statistical interval of 0.02.

### 4.3. Effect of Respiratory Depth on Deposition and Distribution of Microparticles

The depth of respiration describes the volume change of the alveolus during a single respiratory cycle. We change the inlet pressure to adjust the alveolar expansion (contraction) rate and define the alveolar deformation rate by the equation:(34)$$\;\eta_{b}=\frac{\Delta s_{p}}{s_{p}}\times 100\%$$
where $\Delta s_{p}$ is the change of alveolar volume (alveolar area in the two-dimensional (2D) case), which is calculated by the equation:(35)$$\Delta s_{p}=s_{p,t}-s_{p}$$
where $s_{p,t}$ is the alveolar volume at t time and $s_{p}$ is the equilibrium volume (area in 2D case) of the same internal and external alveolar pressure. The increase in the alveolar deformation rate represents the enhancement of respiratory depth. We set the variation range of alveolar deformation rate during normal respiration as (0~11.97%), and deep respiration as (0~29.43%).

We use microparticles with a diameter *D* of 4 μm to simulate the deposition progress, of which the distribution state of initial position at *t* = 0 is shown in Figure 7. In order to study the movement and distribution of microparticles in different gravity conditions, we use *θ* to represent the direction of gravity, which is set as 0°, 30°, 60° and 90°. Different gravity directions simulate various body position changes, which lead to direction changes between acinar and gravity. After 10 respiratory cycles, most of the microparticles are close to the alveolar wall, and we calculate the microparticle distribution and deposition (Figure 8).

In Figure 8a–d, Case 1 represents the process of the natural sedimentation of the microparticles, when no alveolar movement (no respiration) occurs, which is shown as the basic reference. As we can see in Figure 8a, more microparticles distribute at the inlet of alveolus when not breathing. Case 2 represents the normal respiratory process, and more microparticles distribute in the middle alveolus compared with Case 1. Case 3 reveals the deep respiratory process, in which more microparticles distribute to the distal alveolus. We find that, under the same gravity condition, the higher the alveolar deformation rate is, the more microparticles distribute in the distal alveolar wall. It suggests that deep respiration is conducive to the movement and deposition of microparticles in the deeper region of pulmonary acini. Figure 8b–d shows the effect of gravity. In Case 3, microparticles distribute mostly in the deep area of pulmonary acini, followed by Case 1 and Case 2. Therefore, the results of this part of the study suggest that deep breath facilitates the migration and deposition of microparticles to the deeper area in pulmonary acini.

### 4.4. Effect of Microparticle Diameter on Deposition and Distribution of Microparticles

The diameter of microparticles affects their movement and distribution by affecting gravity and its morphology. We use microparticles with diameters *D* of 2, 3, 4, and 5 μm to simulate the deposition progress, of which the distribution state of initial position at t = 0 is shown in Figure 7. We observe the effect of different microparticle diameters on their deposition during ten deep respiration cycles. Similarly, we use *θ* to represent the gravity direction, which is set as 0°, 30°, 60°and 90°. We calculate the microparticle distribution and deposition in Figure 9.

The distribution of microparticles with different diameters in acini is shown in Figure 9a when gravity direction *θ* is 0°. We can observe that fewer microparticles of 2 μm distribute in the middle alveolus. The distribution of 3 μm microparticles is similar to those of 2 μm, yet the number of microparticles is larger. Some microparticles of 4 μm and 5 μm distribute near the distal wall of pulmonary acini. In general, larger particles are more likely to deposit in the alveolar wall at the same respiratory cycle and depth. The results under four gravity conditions (Figure 9a–d) reveal that the direction of gravity is the main reason that affects the deposition location of microparticles, which is more obviously observed in larger microparticles. Our results, therefore, indicate that the pulmonary spatial orientation and the diameter of microparticles strongly impact the efficacy of atomization drug delivery.

### 4.5. Effect of Respiratory Rate on Deposition and Distribution of Microparticles

The respiratory rate describes the rate of volume change of the alveolus during a single respiratory cycle. We set the normal rate in the deep respiratory process as $U_{r}$. In the slow respiratory process, the respiratory rate is 0.5$U_{r}$, while the respiratory rate is 2$U_{r}$ in the rapid process. The distribution state of the initial position at *t* = 0 is shown in Figure 7. We use microparticles with a diameter *D* of 4 μm to simulate the deposition progress, and we observe the effect of different respiratory rates and different gravity directions on microparticle deposition in a deep breath during ten respiratory cycles. The results are shown in Figure 10.

The distribution of microparticles in acini during different respiratory rates is shown in Figure 10a when gravity direction *θ* is set to 0°. It can be observed that, within the normal range of respiratory rates, the rate has no obvious effect on the microparticle deposition or distribution in pulmonary acini, and the microparticle deposition pattern almost remains unchanged. The experimental results under four gravity directions show that the distribution of microparticles in acini is approximately the same. Therefore, we believe that respiration rate has relatively little influence on distribution and deposition in pulmonary acini.

In low Reynolds number cases, the increase in respiratory rate does not affect the flow in pulmonary acini obviously, but only changes the flow velocity. This change will cause some effect on the inertia of particles. However, for particles of 2–5 μm, the influence is limited enough.

## 5. Conclusions and Discussion

In this study, we used IB–LBM to simulate the movement and deposition of solid (liquid or hydrogel) particles numerically in pulmonary acini. We proposed a method based on the elastic mesh support outside the flow field to propose the elastic lung tissue to support the alveolus, and to simulate the alveolar movement during respiration by changing the pressure at the alveolar entrance to promote alveolus expansion and contraction. On this basis, we studied the movement and distribution law of microparticle populations with different microparticle diameters, respiratory depths, respiration rates, and gravity directions. The main conclusions are as follows:(1)Large breathing depth is conducive to the movement and deposition of microparticles in pulmonary acini.(2)The microparticles with a larger diameter are easier to deposit after entering the alveoli.(3)The direction of gravity has a powerful effect on the deposition location of the microparticle.(4)In the same respiratory depth and gravity direction, the respiratory rate has little effect on the distribution of microparticles.


The research conclusions above provide a certain reference for designing aerosol delivery and the reduction in particle pollution exposure. For example, in the design of aerosol delivery, to achieve the best efficacy of an aerosol, the number of drug particles should be maximized in alveoli. To evenly distribute the microparticles, deep breath and change in body position should be recommended when administrating aerosol. In addition, to limit the impacts of haze pollution on health, accelerating respiratory rate is more effective than deep respiration for inhibiting the deposition of microparticles in the lungs.

There are also some limitations to this study. First, we used a two-dimensional model to propose pulmonary acini and microparticle models. However, the distribution and deposition of microparticles are three-dimensional, so what the model reveals is from a limited perspective. Second, this model is limited by the grid resolution; thus, it does not provide the possibility of studying microparticles with tiny diameters (less than 1 μm), which are liable to enter the pulmonary acini. Therefore, to deepen the corresponding research, the current model needs to be further improved.

## Figures and Tables

**Figure 1 micromachines-12-00957-f001:**
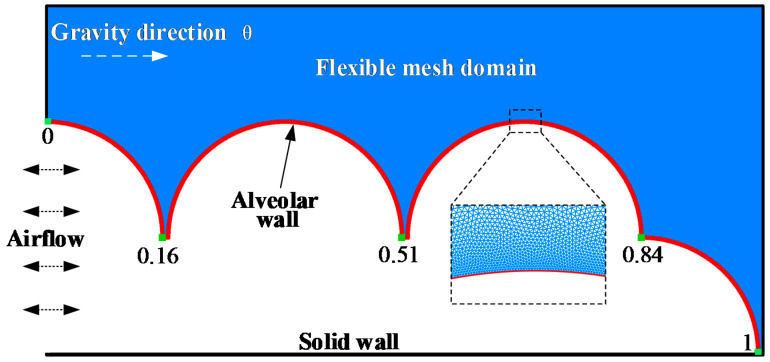
Terminal pulmonary acini.

**Figure 2 micromachines-12-00957-f002:**
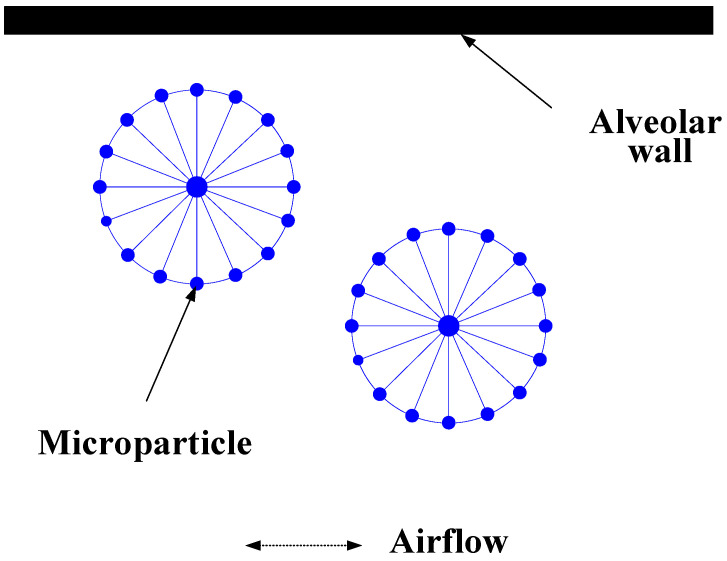
Solid particles model.

**Figure 3 micromachines-12-00957-f003:**
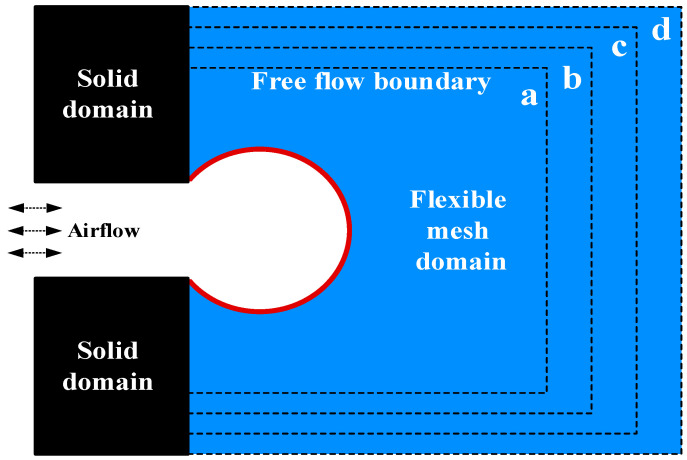
Grid independence verification model.

**Figure 4 micromachines-12-00957-f004:**
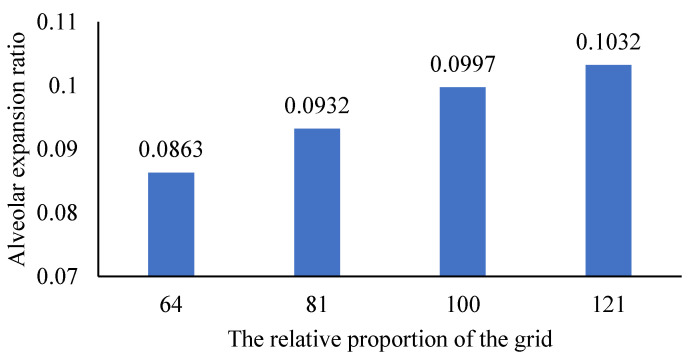
The influence of changing the proportion of the grid on the proportion of alveolar expansion.

**Figure 5 micromachines-12-00957-f005:**
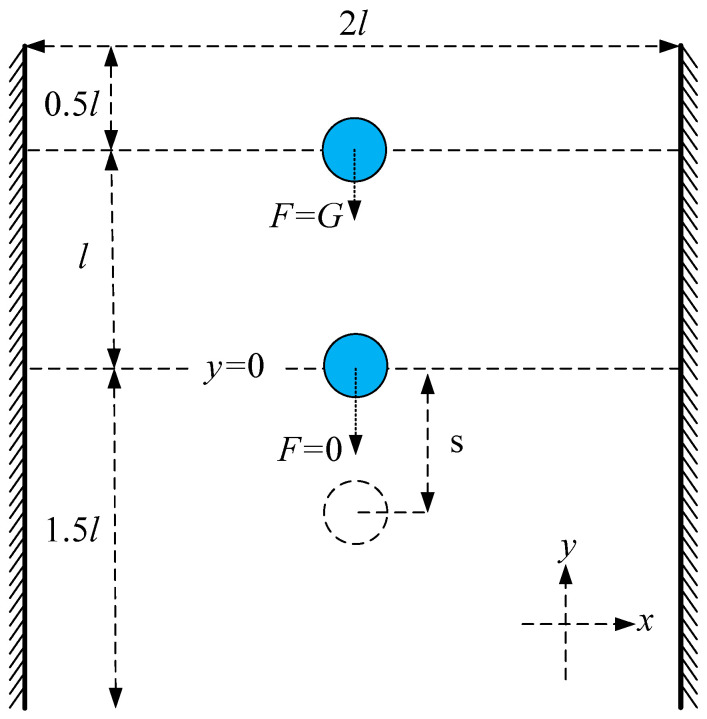
Inertial force validation model.

**Figure 6 micromachines-12-00957-f006:**
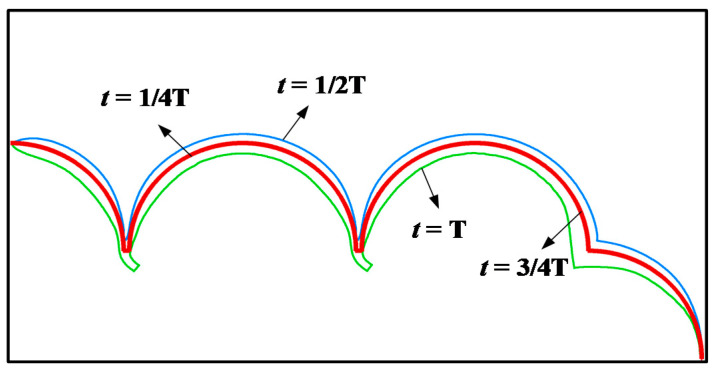
Process of pulmonary acini respiration.

**Figure 7 micromachines-12-00957-f007:**
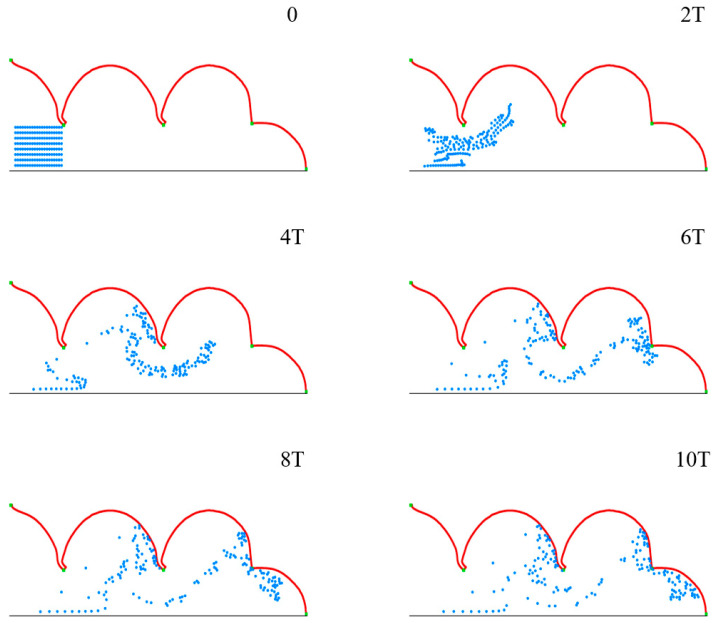
The movement and distribution of microparticles in pulmonary acini during the respiratory cycle.

**Figure 8 micromachines-12-00957-f008:**
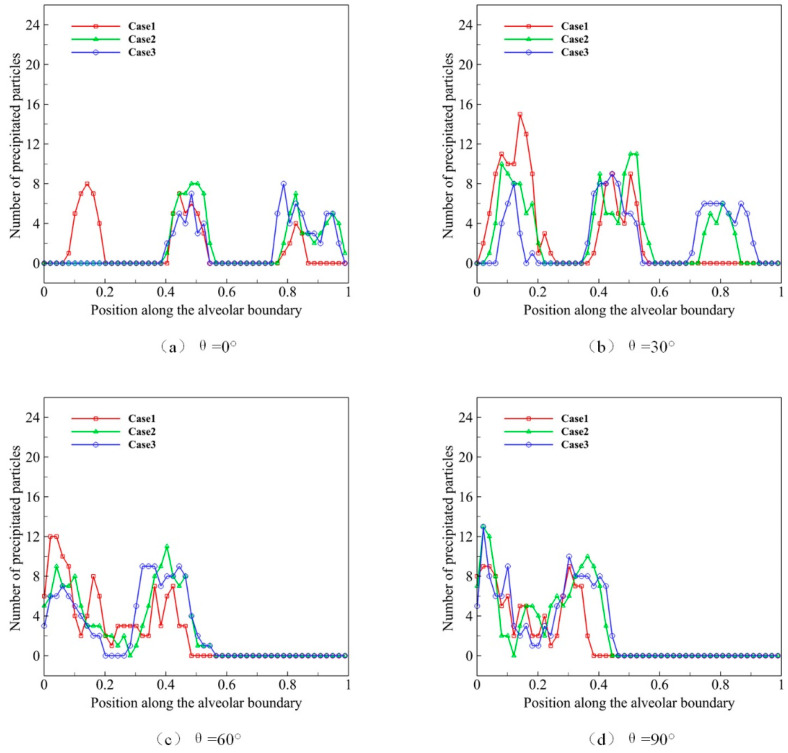
The deposition and distribution of microparticles in pulmonary acini at different respiratory depths.

**Figure 9 micromachines-12-00957-f009:**
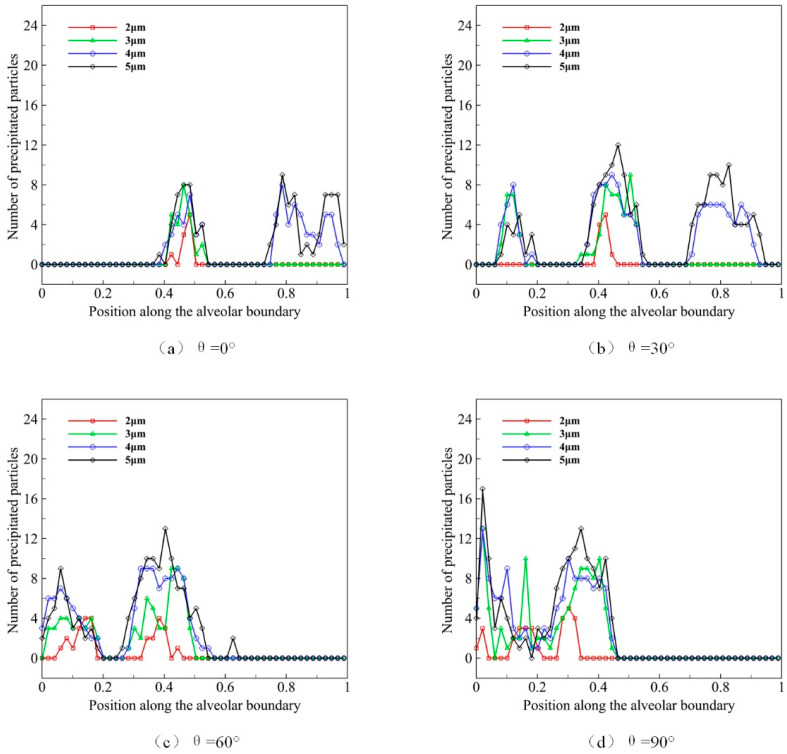
The deposition and distribution of microparticles in pulmonary acini at different microparticle diameters.

**Figure 10 micromachines-12-00957-f010:**
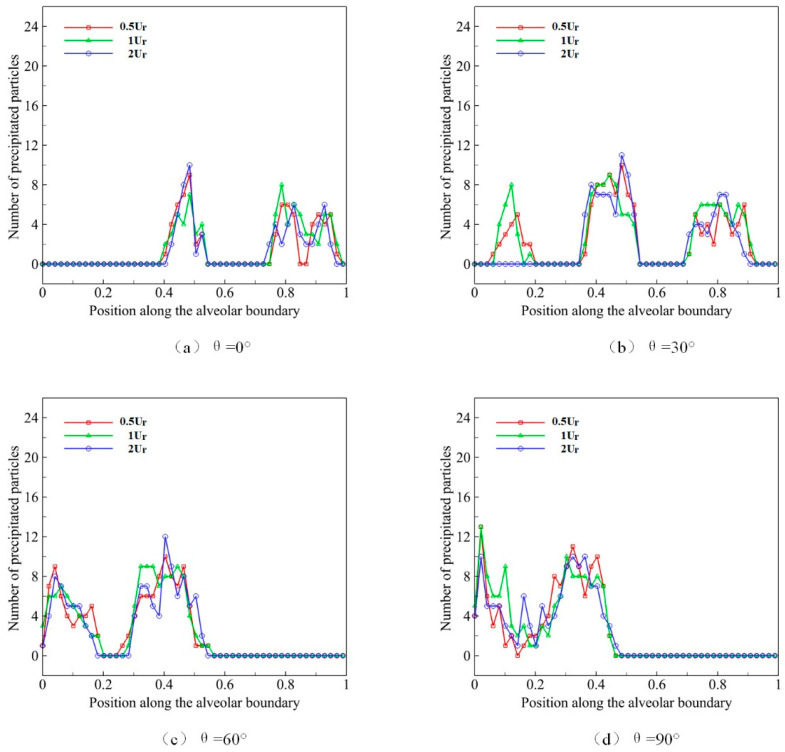
The deposition and distribution of microparticles in pulmonary acini at different respiratory rates.

**Table 1 micromachines-12-00957-t001:** Sliding distance of different microparticle sizes.

*D*	*s*
3 μm	2.22 μm
4 μm	5.66 μm
5 μm	11.37 μm

**Table 2 micromachines-12-00957-t002:** Sliding distance of different inertial force coefficients.

$m_{s}$	*s*
0	5.66 μm
50	11.09 μm
100	16.08 μm

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
