# Peer review of "Study on Deposition Characteristics of Microparticles in Terminal Pulmonary Acini by IB–LBM"

_micromachines, 2021, doi:10.3390/mi12080957_

Round 1
Reviewer 1 Report
The authors apply an IB-LB method to study the deposition of airborne microparticles in terminal pulmonary acini. A number of parameters are considered to quantify the deposition characteristics, which include the gravity direction, respiratory rate, and particle size, etc. The authors show that deposition is highly dependent on gravity direction and particle size. The respiratory rate was shown to have little effect on the deposition rate and site.
1. The introduction needs to review the drug delivery studies in the human body in general before the authors narrow their focus to drug delivery in pulmonary systems. In particular, substantial progress has been made in recent years regarding nanoparticle and microparticle delivery in blood circulation (e.g., Lee et al. 2013, 10.1038/srep02079; Muller et al. 2014, 10.1038/srep04871; Liu et al. 2018, 10.1016/j.compfluid.2018.03.022).
2. In the method section, the recent development of the LB-LD method by Liu et al. (2019) (10.1002/fld.4752) that is capable of simulating nano-to-microscale particle transport (10.1063/1.5110604) suspended in LB fluids deserves to be discussed and compared with the author's IB-LB method for microparticle transport.
3. The "validation" section certainly is more of a qualitative method verification. Validation means more quantitative comparison with experimental or theoretical results. I suggest the authors change their claim to numerical verification instead.
4. The motion of the pulmonary acini seems to be very small. Are these prescribed motions clinically relevant? Could the authors clarify how they specify the motions and what literature they base off of?
5. Can the authors clarify how they define the gravitation direction?
6. Particle size effect. Although the authors focus on microparticle deposition, could they comment on how nanoscale particulate (e.g., pm 2.5 airborne particles are nanoscale) would behave in the system?
7. Could the authors provide some physical insights on why the resp rate has little effect on the deposition characteristics?
Author Response
Dear Reviewer,
Thank you very much for your professional and constructive comments on our paper. The responses are included in the attached file!
Best wishes!

Reviewer 2 Report
The present paper aims at employing a LB-IBM approach to reproduce the respiratory movement of terminal pulmonary acini.
The work is interesting and thoroughly done and the topic is worth to be investigated.
First of all, I suggest the Authors to carefully check the grammar in the manuscript.
Below some points:
1) abstract: "in this work we establish..."
Perhaps is better to use we propose or we employ. Anyway, as suggested above, please check the whole manuscript.
2) Why the choice of a MRT model instead of a simpler BGK or Regularized approach (see [1-2]).
[1] The lattice Boltzmann equation: theory and applications
R Benzi, S Succi, M Vergassola, Physics Reports 222 (3), 145-197, 1992
[2] Lattice Boltzmann method with regularized pre-collision distribution functions
J Latt, B Chopard, Mathematics and Computers in Simulation 72 (2-6), 165-168,2006
[3] Regularized lattice Bhatnagar-Gross-Krook model for two- and three-dimensional cavity flow simulations, A. Montessori, G. Falcucci, P. Prestininzi, M. La Rocca, and S. Succi, Phys. Rev. E 2014.
[4] Reassessing the single relaxation time Lattice Boltzmann method for the simulation of Darcy’s flows, Prestininzi et al., Int J. of Mod. Phys. C, 2014
[5] The lattice Boltzmann equation: for fluid dynamics and beyond
S Succi ,Oxford university press,2001
3) The Authors are encouraged to take a look and take into consideration the following relevant works dealing with the use of LB for flows in complex geometries.
[1] Three-dimensional flows in complex geometries with the lattice Boltzmann method
S Succi, E Foti, F Higuera, EPL (Europhysics Letters) 10 (5), 433, 1989
[2] Lattice Boltzmann appraoch for complex nonequilibrium flows, Montessori et al. PRE, 2015
[3]Mesoscale modelling of near-contact interactions for complex flowing interfaces, Montessori et al., Journal of Fluid Mechanics 2019
[4] Tölke, Jonas. "Lattice Boltzmann simulations of binary fluid flow through porous media." Philosophical Transactions of the Royal Society of London. Series A: Mathematical, Physical and Engineering Sciences 360.1792 (2002): 535-545.
[5] Extended friction elucidates the breakdown of fast water transport in graphene oxide membranes, Montessori et al., EPL (Europhysics Letters) 116 (5), 54002
Author Response
Dear Reviewer,
Thank you very much for your professional and constructive comments on our paper. The responses are included in the attached file.
Best wishes!
